# Combined Experimental and Theoretical Investigation on Formation of Size-Controlled Silver Nanoclusters under Gas Phase

**DOI:** 10.3390/bios12050282

**Published:** 2022-04-28

**Authors:** Chuhang Zhang

**Affiliations:** Physics Department, Zhejiang University of Science and Technology, 318 Liuhe Road, Xihu District, Hangzhou 310008, China; 114061@zust.edu.cn; Tel.: +86-13750865421

**Keywords:** silver nanocluster, formation mechanism, Smoluchowski equation, fragmentation

## Abstract

Metallic nanoclusters (NCs) have been predicted to achieve the best Surface-Enhanced Raman Scattering (SERS) due to the controllable amount of atoms and structures in NCs. The Local Surface Plasmon Resonance (LSPR) effect on silver metal NCs (Ag${}_{n}$) enables it to be a promising candidate for manipulating the LSPR peak by controlling the size of NCs, which in turn demands a full understanding of the formation mechanism of Ag${}_{n}$. Here, we apply an extended Smoluchowski rate equation coupled with a fragmentation scheme to investigate the growth of size-selected silver NCs generated via a modulated pulsed power magnetron sputtering (MPP-MSP). A temperature-dependent fragmentation coefficient *D* is proposed and integrated into the rate equations. The consistency between the computational and experimental results shows that in relative low peak power ($P_{p}\leq 800$ W), the recombination of cation and anion species are the dominant mechanism for NC growth. However, in the higher $P_{p}$ region (≥800 W), the fragmentation mechanism becomes more impactful, leading to the formation of smaller NCs. The scanning electron microscopy observation shows the Ag${}_{36}$ is successfully soft-landed and immobilized on a strontium titanate crystal, which facilitates the application of the Ag${}_{n}$/STO to the SERS research.

## 1. Introduction

The fabrication of size-controlled nanoclusters (NCs) is of significant importance to research fields such as photocatalysis, electrocatalysis, energy and biosensing [1,2,3,4,5,6,7,8,9,10]. Due to the size-specific properties, noble metal NCs such as silver NCs show a localized surface plasmon resonance (LSPR) effect, which in turn leads to surface-enhanced Raman scattering (SERS) [11,12,13,14]. It is also predicted that the structure and size of the NCs will cause the shift in the LSPR peaks [15,16,17,18]. This opens up a possible way to achieve the best SERS enhancement by manipulating the physical properties of the NCs such as size and structure. Recently, Ye et al. reported an improved SERS in a gold nanostructure system fabricated on a liquid surface [19]. In order to synthesize intensive and size-controllable NCs, a complete understanding of the formation mechanism of NC is always required.

In our previous study, an NC generation source based on a modulated pulsed power magnetron sputtering (MPP-MSP) technique was developed [20]. Pulsed power was utilized to provide dense sputtered materials, which in turn created an intense ion current of NCs with an intensity up to 5 nA. Coupled with a quadrupole mass spectrometer (Q-MS), silver NCs (Ag${}_{n}$) of sizes from n=1−100 were generated and then size-selected for deposition on a SrTiO${}_{3}$ crystal surface. By controlling parameters of peak power ($P_{p}$) and repetition rate (*f*) of the pulsed power, the size of Ag${}_{n}$ were well controlled [20].

In order to understand the formation mechanism of the NCs prepared by the MPP-MSP source, we have extended the conventional Smoluchowski rate Equation [21] by considering the cation–anion recombination during condensation [22]. The calculation results by extended Smoluchowski rate equations were in good consistence with the mass spectra obtained by Q-MS, indicating that a sequential growth mechanism occurs under low peak power and repetition rate. Furthermore, the extended equation has successfully predicted the formation of large neutral clusters under relatively high $P_{p}$ (up to 700 W) and *f* [22]. However, as $P_{p}$ further increases to higher values, both the size and intensity of ionic NCs are dramatically reduced. According to the extended equation, high $P_{p}$ leads to denser ionic species, which in turn enhances the cation–anion combination. The behavior that the size of NCs is reduced under high $P_{p}$ cannot be explained by the extended Smoluchowski model and was proposed to be attributed to the possible fragmentation of large NCs under high plasma temperatures. In order to realize full controllability of relative large NCs, the extended Smulochowski rate equations should be further modified.

In this study, we integrate the fragmentation mechanism of large NCs into the extended Smoluchowski equation to investigate the formation mechanism of silver NC generated by MPP-MSP. By controlling the ionization fraction, sputtered material density and fragmentation coefficient, the formation of Ag NCs under high peak power is simulated. The calculated results are also compared with the mass spectra obtained by Q-MS. The consistency between the calculation and experiment suggests that large NCs fragment into small ones under a high NC temperature, which is caused by high $P_{p}$. Using a soft-landing technique, the size-selected silver NCs (Ag${}_{n}$) are deposited and immobilized on a strontium titanate (STO) crystal, which facilitates the future study of the SERS performance of Ag${}_{n}$/STO system.

## 2. Materials and Methods

### 2.1. Experiment

Figure 1 shows the sketch of the experimental setup, which includes a magnetron sputtering source powered by a modulated pulsed power (MPP), a condensation cell, an octopole ion guide (OPIG), an ion bender, a quadruple mass spectrometer (Q-MS; Extrel, MAX-16000) and a sample platform [20,22]. The 99.99% pure silver target (Good Fellow Co., Inc., Shanghai, China) is mounted on the sputtering anode as the target, and the peak power ($P_{p}$) and the repetition rate (*f*) of the MPP were manipulated. Typical values for the peak voltage, discharge current and peak power fell between −320 and −500 V; 1.2 and 3.0 A; and 400 and 1500 W, respectively. It should be noted that the initial position is 0.05 m in front of the target where we set *x* as 0. Argon and helium flows into the cell to facilitate the sputtering process, and the typical Ar and He flow rates were from 100 and 400 sccm, respectively, leading to a background pressure of approximately 20 Pa. The sputtered materials including silver atoms, anions and cations are cooled down and aggregate to NCs in the condensation cell, which is 290 mm long. After that, the silver NCs are guided by the OPIG to the ion bender where charged species are deflected to Q-MS. The sizes of the Ag NCs are analyzed, and mass spectra are acquired in Q-MS. After size-selection, Ag NCs with specific sizes are deposited onto the strontium titanate crystal (STO (110), MTI Co., Inc., Richmond, CA, USA), which is pre-washed by a mixed solution of 7 mL of NH${}_{4}$, 3 mL of HF and 10 mL of deionized water. It should be mentioned that there is a movable shield mounting between the outlet of the Q-MS and the sample platform to ensure no NCs will be deposited before the size-selection process in Q-MS is completed. In order to reduce the fragmentation of NCs during deposition, a retarding potential of −1.5 V is applied to the STO. In order to measure the temperature (*T*, in Kelvin degree) of the NCs, a Langmuir probe is placed at different locations in front of the target in the condensation cell.

When the deposition process starts, the shield is opened, and the size-selected Ag${}_{n}$ are deposited onto the STO, and the deposition amount is estimated as the product of the deposition time and the apparent ion current in the mass spectrum for a specific Ag${}_{n}$. The diameter of the outlet of the Q-MS is around 0.3 cm, leading to an apparent deposition area of $7\times 10^{-2}$ cm${}^{2}$ on STO. After deposition, immediate scanning electron microscopy (ZEISS SUPRA 55 SAPPHIRE, Carl Zeiss, Shanghai, China) observation is performed on the Ag${}_{n}$/STO sample under vacuum conditions. The size distribution of the Ag${}_{n}$ is analyzed by an Igor Pro 8.0 software (WaveMetrics, Lake Oswego, OR, USA).

### 2.2. Simulation

For a computational model, we adapt the extended Smoluchowski model reported in our previous report, in which the aggregation rate equations for the Extended Smoluchowski model are given by [22]
(1)$$\begin{array}{cc} \frac{dn_{k}}{dt} & =\frac{1}{2}K_{i,k-i}\sum_{i=1}^{k-1}n_{i}n_{k-i}-K_{k,i}\sum_{i=1}^{\infty }n_{k}n_{i}\\ \; & -K_{k,i}^{{}^{\prime }}\sum_{i=1}^{\infty }n_{k}\left(m_{i}^{+}+m_{i}^{-}\right)+K_{i,k-i}^{{}^{\prime \prime }}\sum_{i=1}^{k-1}m_{i}^{+}m_{k-i}^{-}, \end{array}$$
(2)$$\frac{dm_{k}^{+}}{dt}=K_{k-i,i}^{{}^{\prime }}\sum_{i=1}^{k-1}m_{k-i}^{+}n_{i}-K_{k,i}^{{}^{\prime }}\sum_{i=1}^{\infty }m_{k}^{+}n_{i}-K_{k,i}^{{}^{\prime \prime }}\sum_{i=1}^{\infty }m_{k}^{+}m_{i}^{-},$$
(3)$$\frac{dm_{k}^{-}}{dt}=K_{k-i,i}^{{}^{\prime }}\sum_{i=1}^{k-1}m_{k-i}^{-}n_{i}-K_{k,i}^{{}^{\prime }}\sum_{i=1}^{\infty }m_{k}^{-}n_{i}-K_{k,i}^{{}^{\prime \prime }}\sum_{i=1}^{\infty }m_{k}^{-}m_{i}^{+},$$
where $n_{k}$, $m_{k}^{-}$ and $m_{k}^{+}$ represent the number of neutral, anionic and cationic NCs with size *k*. The detailed descriptions for each term in the above rate equations have been discussed in our previous report [22]. In this research, we will follow the parameters such as initial sputtered material density $n_{0}$, ionization fraction σ, polarizability *a* and constant *u*, which are already determined in our previous report [22].

Based on Equation (Equation 1), we now consider the effect of the possible NC fragmentation mechanism. In order to simplify the model, we have two assumptions: (1) the fragmentation probability of a specific cluster is dependent on the difference between the binding energies of mother and daughter clusters; (2) cluster anions, cations and neutrals with the same size have the same fragmentation probability under the same temperature. For binding energies of the silver NCs, we refer to the previous publications [23,24,25]. Based on these, the fragmentation coefficient of Ag${}_{i}$ to Ag${}_{k}$ and Ag${}_{i-k}$ under temperature *T*, namely $D_{i,k}\left(▵E,T\right)$, is proposed to be
(4)$$D_{i,k}\left(▵E,T\right)=\frac{1}{1+e^{-▵E/kT}}$$
where ▵E is the binding energy difference between a mother cluster Ag${}_{i}$ and two daughter clusters Ag${}_{k}$ and Ag${}_{i-k}$. *k* and *T* are the Boltzmann’s constant and NC temperature measured in Kelvin, respectively. Based on Equation (Equation 4), ▵E<0 will lead to a small value of *D*, which means if the binding energy of a mother cluster is less than the binding energies of two daughter clusters, the fragmentation of this mother cluster is energetically unfavorable. In this case, raising the temperature of NCs will increase the value of *D*, which means a higher NC temperature will cause higher fragmentation probability. For example, the fragmentation coefficient of a Ag${}_{12}$ fragment to Ag${}_{3}$ and Ag${}_{9}$ when T=1000 K is around $D_{12,3}\left(-0.1,1000\right)=\frac{1}{1+e^{0.1/0.087}}=24\%$. *D* should have a unit of s${}^{-1}$ because it denotes the change in the population per unit time. The integration of Equation (Equation 4) to Equation (Equation 1) will lead to
(5)$$\begin{array}{cc} \frac{dn_{k}}{dt} & =\frac{1}{2}K_{i,k-i}\sum_{i=1}^{k-1}n_{i}n_{k-i}-K_{k,i}\sum_{i=1}^{\infty }n_{k}n_{i}\\ & -K_{k,i}^{{}^{\prime }}\sum_{i=1}^{\infty }n_{k}\left(m_{i}^{+}+m_{i}^{-}\right)-\sum_{j=1}^{k-1}D_{k,j}\left(▵E,T\right)n_{k}+\frac{1}{2}K_{i,k-i}^{{}^{\prime \prime }}\sum_{i=1}^{k-1}m_{i}^{-}m_{k-i}^{+}\\ & +\sum_{i=k+1}^{\infty }D_{i,k}\left(▵E,T\right)n_{i}+\sum_{i=k+1}^{\infty }D_{i,k}\left(▵E,T\right)m_{i}^{+}+\sum_{i=k+1}^{\infty }D_{i,k}\left(▵E,T\right)m_{i}^{-}. \end{array}$$

Compared to Equation (Equation 1), there are four additional terms, i.e., the fourth, sixth, seventh and last terms. The fourth term means the fragmentation of neutral NCs of size *k* to smaller daughter NCs. The sixth, seventh and last terms denote the formation of neutral NCs of size *k* due to the fragmentation of larger neutral, cationic and anionic NCs.

The kinetic equations for NC cations and anions are thus expressed as: (6)$$\begin{array}{cc} \frac{dm_{k}^{+}}{dt} & =K_{k-i,i}^{{}^{\prime }}\sum_{i=1}^{k-1}m_{k-i}^{+}n_{i}-K_{k,i}^{{}^{\prime }}\sum_{i=1}^{\infty }m_{k}^{+}n_{i}\\ & -K_{k,i}^{{}^{\prime \prime }}\sum_{i=1}^{\infty }m_{k}^{+}m_{i}^{-}-\sum_{j=1}^{k-1}D_{k,j}\left(▵E,T\right)m_{k}^{+}+\sum_{i=k+1}^{\infty }D_{i,k}\left(▵E,T\right)m_{i}^{+}, \end{array}$$
(7)$$\begin{array}{cc} \frac{dm_{k}^{-}}{dt} & =K_{k-i,i}^{{}^{\prime }}\sum_{i=1}^{k-1}m_{k-i}^{-}n_{i}-K_{k,i}^{{}^{\prime }}\sum_{i=1}^{\infty }m_{k}^{-}n_{i}\\ & -K_{k,i}^{{}^{\prime \prime }}\sum_{i=1}^{\infty }m_{k}^{-}m_{i}^{+}-\sum_{j=1}^{k-1}D_{k,j}\left(▵E,T\right)m_{k}^{-}+\sum_{i=k+1}^{\infty }D_{i,k}\left(▵E,T\right)m_{i}^{-}. \end{array}$$

Compared to Equations (Equation 2) and (Equation 3), the fourth and the last terms in Equations (Equation 6) and (Equation 7) are due to the fragmentation mechanism. The fourth term denotes the annihilation of NCs of size *k* via fragmentation to smaller NCs. The fifth term is the formation of NCs via fragmentation of larger NC ions. Because the traveling time for the sputtered materials in the condensation cell with a length of 290 mm is around 40 ms [20], the simulation time interval dt is $10^{-5}$ s and the simulation steps for the whole aggregation process is around 4000.

## 3. Results and Discssion

Figure 2 shows dependent behaviors of nanocluster temperature (*T*) and probe position (*x*) under different peak power values. It can be clearly seen that *T* drops exponentially with *x* in the condensation cell. By fitting the curves by exponential decay curves, we find the function for the best fit curve are (a) $810e^{-3.12x}$, (b) $900e^{-3.21x}$, (c) $1050e^{-3.06x}$ and (d) $1200e^{-3.11x}$, respectively. Therefore, we can roughly conclude that the relation between T,x and $P_{p}$ is $T=680\left(1+P_{p}/1600\right)e^{-3x}$.

Figure 3a shows the mass spectra of Ag anions obtained under varied $P_{p}$ from 400 W to 1100 W under the setting of ▵E→−∞, which means no fragmentation mechanism of NCs is considered at this stage. It should be mentioned that a Matlab program is utilized to pick up all peaks in the original mass spectrum. It can be clearly seen that the maximum intensity reaches around 5 nA, and as $P_{p}$ increases from 400 W to 600 W, the size distribution of the spectrum shifts to a larger-sized region, indicating the formation of larger-sized NCs. However, as $P_{p}$ further increases to 800 W and 1100 W, the smaller-sized NCs become more abundant, and the total intensity of the spectra is dramatically reduced. These observations are generally consistent with the results obtained by the model, as shown in Figure 3b. However, a close look at Figure 3a,b at $P_{p}=800$ W (red color) indicates that the the size of the most abundant size slightly shifts from 12 to 7-mer, as shown in Figure 3a,b. The observation shown in Figure 3 suggests that although the recombination of anion–cation species at moderate $P_{p}<800$ W is the dominant reason for the size shifting, fragmentation of larger NC ions may occur when $P_{p}>800$ W. From the inset, we can clearly see that the maximum ion current for Ag${}_{n}$ reaches 5 nA at n=24.

If cluster fragmentation occurs as $P_{p}$ increases, this mechanism will be more obvious for a neutral cluster because neutral clusters do not undergo cation–anion aggregation, as indicated in Equations (Equation 2) and (Equation 3). Figure 4 show the mass spectra of neutral species obtained under the same conditions as in the anion case where ▵E→∞, i.e., no fragmentation mechanism is considered. When $P_{p}$ = 400 W and 600 W, we can see the mass spectrum and the size distributions of NCs obtained by Q-MS and the model are generally consistent with each other. As $P_{p}$ increases up to 800 W, although the size distribution obtained by the Q-MS and the model are similar, the smaller-sized NCs in Figure 4a are clearly more abundant than in Figure 4b. As $P_{p}$ further increases up to 1100 W, a clear deviation in the size distributions between Figure 4a,b can be observed: for mass spectrum obtained by Q-MS, smaller-sized NCs below 40-mer become more abundant, while in the result obtained by the model, lager sized NCs above 60-mer are still more abundant species. On the other hand, the total intensity for NC neutrals for $P_{p}$ = 800 W and 1100 W are around 2800 a.u. and 4200 a.u., respectively.

The discrepancy shown in Figure 4a,b suggested that large NCs may fragment to smaller ones at high $P_{p}$ along with the recombination of anion–cation species. This proposal is based on the consideration that if the recombination of anion–cation species is the dominant mechanism, both the total intensity and size of NC neutrals will increase, which is not the case, as we can see in Figure 4a. Therefore, we recalculate the size distributions of NC neutrals by following rate Equations (Equation 5)–(Equation 7), and the simulation results are shown below in Figure 5b. In order to determine the fragmentation coefficient $D_{i,k}\left(▵E,T\right)$, we refer to the binding energies for Ag${}_{n}$ reported values in refs. [23,24,25] and the temperature presented in Figure 2. When the initial sputtered materials are set to $n_{0}$ = 4.0 × 10${}^{17}$, σ = 9%, 1.7 × 10${}^{18}$, σ = 12%, $n_{0}$ = 3.5 × 10${}^{18}$, σ = 18% and $n_{0}$ = 6.5 × 10${}^{18}$, σ = 21%, which correspond to $P_{p}$ = 400 W, 600 W, 800 W and 1100 W, the size distributions are quite similar to the mass spectra shown in Figure 5a, indicating the fragmentation mechanism become more dominant at high $P_{p}$.

Comparing the above-calculated and experimental results, an NC deformation mechanism is integrated into the growth for NCs generated via MPP-MSP. When the peak power ($P_{p}$) is below 600 W, the relatively low NC temperature (around 800 K at *x* = 0.05 m) leads to a small fragmentation coefficient. Therefore, the formation mechanism of the NCs is mainly attributed to the conventional neutral–ionic, neutral–neutral and anion–cation species, and the fragmentation of large NCs is not obvious. As $P_{p}$ increases to 800 W, the temperature of the NCs near the target is dramatically increased to around 1000 K, which in turn leads to a fragmentation coefficient according to Equation (Equation 4). In this situation, large NCs intend to fragment to smaller NCs, resulting in the reduction in the intensity of large NCs such as 60-mers and the increase in the smaller NCs such as 20-mers.

After size selection, the Ag${}_{36}$ are soft-landed on a strontium titanate crystal (STO, (110)) by applying a retarding potential of −1.5 V on the sample platform. The deposition current for Ag${}_{36}$ is 2 nA, and the deposition time is 200 s, leading to a total deposition amount of $2.5\times 10^{12}$ NCs. The estimated number density of the Ag${}_{36}$ on STO is therefore around $3.5\times 10^{13}$ cm${}^{-2}$. Figure 6a presents a scanning electron microscopy (SEM) image of Ag${}_{36}$/STO, and Figure 6b is the corresponding size distribution. It can bee seen clearly that almost a monolayer of Ag${}_{36}$ is present on STO, and no obvious overlapping of NCs can be observed. The narrow size distribution shown in Figure 6b indicates that most of the deposited Ag${}_{36}$ are immobilized on STO during the deposition and SEM observation time period. The inset shows the energy dispersive X-ray (EDX) spectrum for Ag${}_{36}$ (deposition amount: 2 nA for 200 s) deposited on a silicone substrate in which a clear silver peak can be observed. For the next step, we will investigate the surface-enhanced Raman scattering for Ag${}_{n}$/STO with different sizes of NCs.

## 4. Conclusions

Size-selected Ag NCs were generated via MPP-MSP, and the formation mechanism was investigated by integrating a fragmentation mechanism to the extended Smoluchowski model. By a Langmuir probe, it is found that the NC temperature goes up when peak power increases. The formation of large NCs under relatively low power is due to the neutral–neutral, neutral–ion and anion–cation species. As peak power increases to 800 W, the fragmentation of large NCs such as 60-mers starts to become overwhelming, which results from the relatively high NC temperature. In this stage, smaller-sized NCs (below 40-mers) become more abundant than the larger ones. By using the retarding potential technique, the size-selected Ag${}_{n}$ soft-land and immobilize on STO. The results of this work may provide guidance for generating intensive size-selected silver NCs and in turn facilitate the research of surface-enhanced Raman scattering on size-selected coinage metal clusters.

## Figures and Tables

**Figure 1 biosensors-12-00282-f001:**
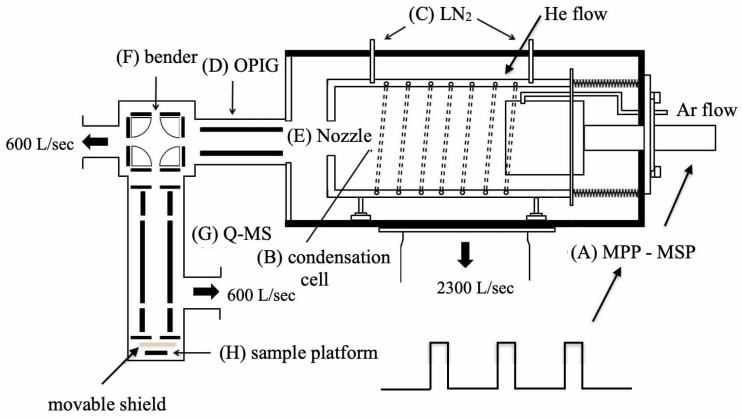
Experimental setup for nanocluster generation based on a modulated pulsed power magnetron sputtering.

**Figure 2 biosensors-12-00282-f002:**
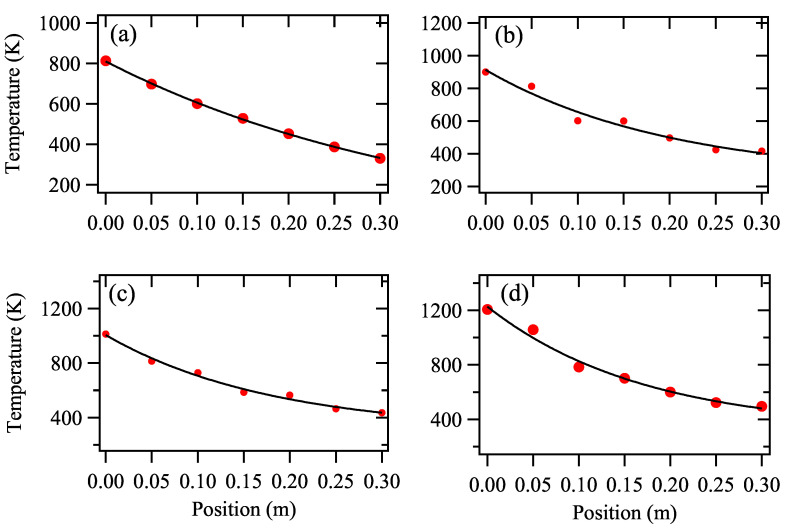
Dependent behavior of nanoclsuters temperature (*T*) and the position of the probe to the target (*x*) measure under $P_{p}$ = (**a**) 400 W, (**b**) 600 W, (**c**) 800 W and (**d**) 1100 W. Repetition rate *f* = 60 Hz. The red dots and the black line represent experimental data and fitting curve, respectively.

**Figure 3 biosensors-12-00282-f003:**
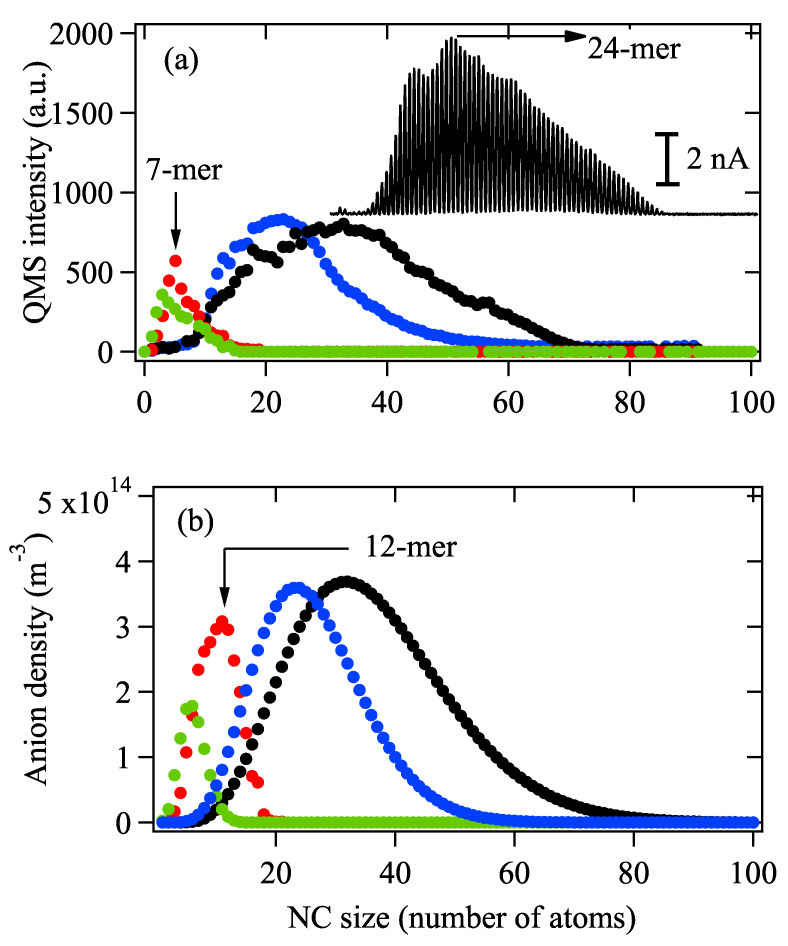
(**a**) Mass spectra of Ag NC anions generated with $P_{p}$ = 400 W (blue), 600 W (black), 800 W (red) and 1100 W (green) at repetition rate *f* = 60 Hz. (**b**) Size distribution of NC anions obtained using the simulation model. Initial sputtered material $n_{0}$ = 4.0 × 10${}^{17}$, σ = 9% (blue), 1.7 × 10${}^{18}$, σ = 12% (black), $n_{0}$ = 3.5 × 10${}^{18}$, σ = 18% (red) and $n_{0}$ = 6.5 × 10${}^{18}$, σ = 21% (green). The inset shows the original mass spectrum for $P_{p}=600$ W (black) acquired by Q-MS. Other parameter set: *u* = 2, *a* = 4.

**Figure 4 biosensors-12-00282-f004:**
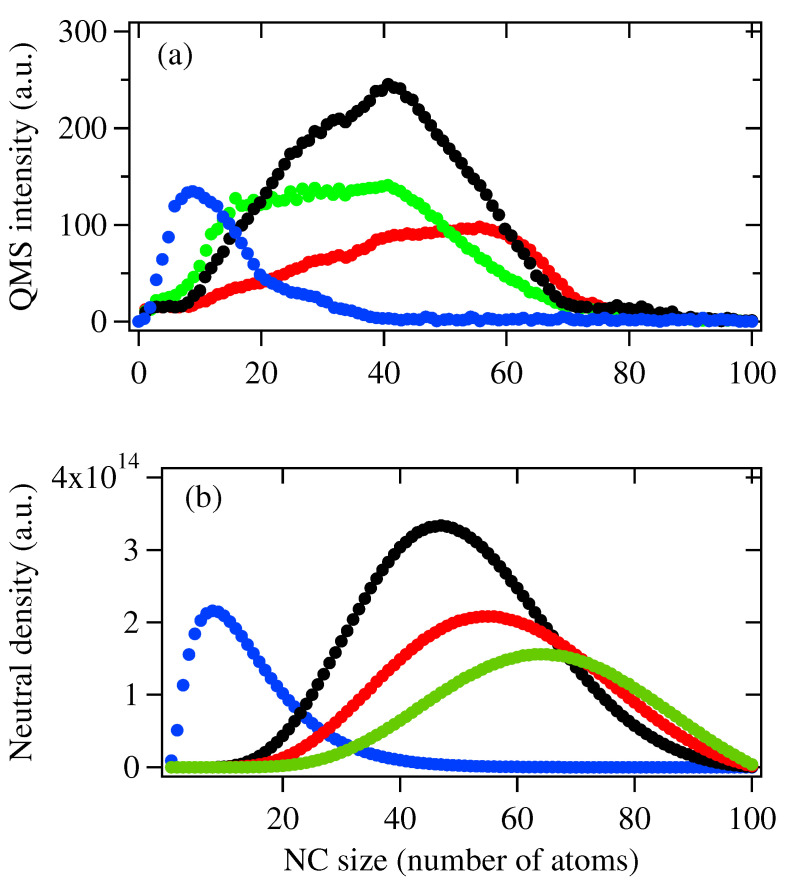
(**a**) Mass spectra of Ag NC neutrals generated with $P_{p}$ = 400 W (blue), 600 W (black) 800 W (red) and 1100 W (green) at repetition rate *f* = 60 Hz. (**b**) Size distribution of NC anions obtained using the simulation model. Initial sputtered material $n_{0}$ = 4.0 × 10${}^{17}$, σ = 9% (blue), 1.7 × 10${}^{18}$, σ = 12% (black), $n_{0}$ = 3.5 × 10${}^{18}$, σ = 18% (red), and $n_{0}$ = 6.5 × 10${}^{18}$, σ = 21% (green). Other parameter set: *u* = 2, *a* = 4.

**Figure 5 biosensors-12-00282-f005:**
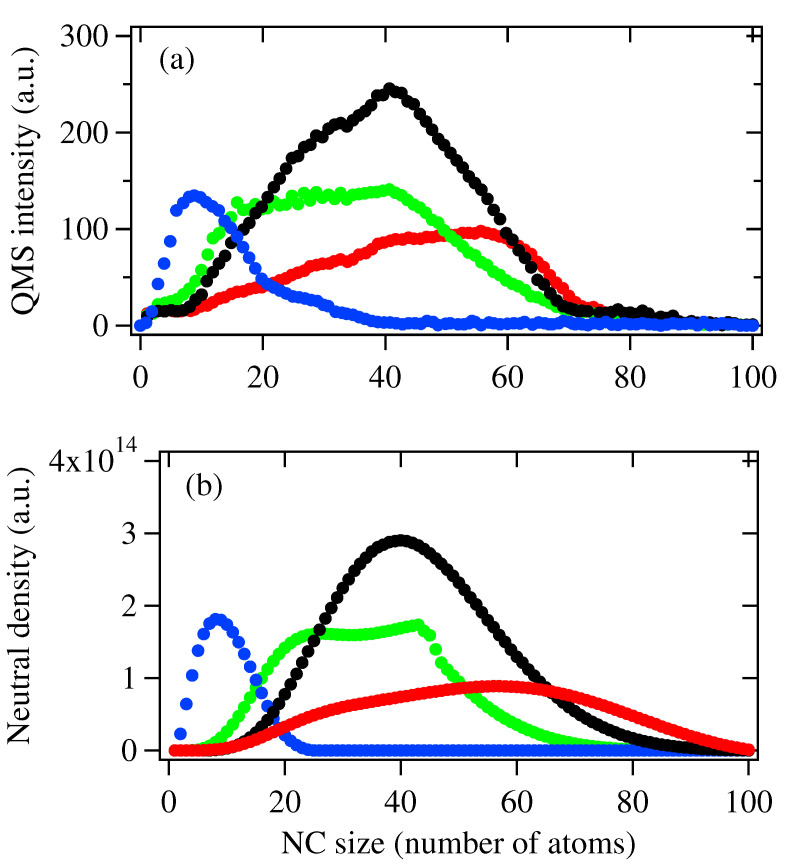
(**a**) Mass spectra of Ag NC neutrals generated with $P_{p}$ = 400 W (blue), 600 W (black), 800 W (red) and 1100 W (green) at repetition rate *f* = 60 Hz. (**b**) Size distribution of NC anions obtained using the simulation model. Initial sputtered material $n_{0}$ = 4.0 × 10${}^{17}$, σ = 9% (blue), 1.7 × 10${}^{18}$, σ = 12% (black), $n_{0}$ = 3.5 × 10${}^{18}$, σ = 18% (red) and $n_{0}$ = 6.5 × 10${}^{18}$, σ = 21% (green). Other parameter set: *u* = 2, *a* = 4.

**Figure 6 biosensors-12-00282-f006:**
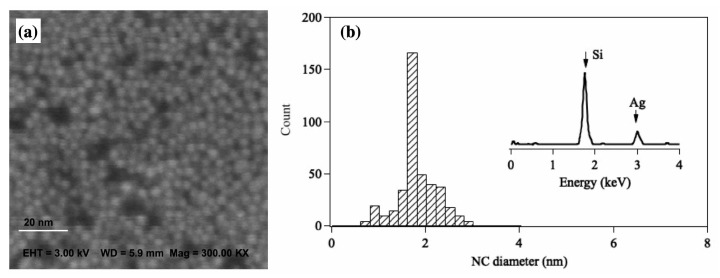
(**a**) SEM image of Ag${}_{36}$/STO. (**b**) Size distribution of Ag${}_{36}$. The inset is the EDX spectrum of Ag${}_{36}$ deposited on a silicone substrate, where the Ag peak can be seen.

## Data Availability

Not applicable.

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
