# Peer review of "Combined Experimental and Theoretical Investigation on Formation of Size-Controlled Silver Nanoclusters under Gas Phase"

_biosensors, 2022, doi:10.3390/bios12050282_

Round 1

Reviewer 1 Report

The manuscript seems to me rather interesting. The author analyzes the controlled formation of Ag nanoclusters. The manuscript presents a solid combined experimental and theoretical study. It does not contain serious drawbacks. In my opinion, the manuscript is well written; the methods are described in detail, which allows the reader to repeat all the results, if necessary. Presented in the manuscript results seem solid, discussion, and main conclusions sound reasonable. On the presentation of the results, I have no critical remarks. Therefore, I think the manuscript can be accepted for publication.

Reviewer 2 Report

The manuscript titled 'Combined experimental and theoretical investigation...' submitted for evaluation to Biosensors and having as single author C. Zhang deals with with details on understanding the formation of silver nanoparticles.

The Abstract (lines 7-13) its a little bit difficult to be understand, therefore it must be revised in order to make more clear. Abstract should summarize the most important aspect of the manuscript.

The Introduction has 12 references on the main subject; however, this subject is of great importance and many important references are skipped. Literature data is abundant in very cent ones, so this issue should also be improved.

Matreials and Method chapter starts directly with Figure 1, I suggest to be moved after the first paragraph and increase also its size.

-line 95-smoluchowski should be written with S;

Results and Discussion chapter starts directly with Figure 2, this should be moved after first paragraph as well.

-line 220- authors should be author.

The work has only 15 references on a subject greatly covered by the literature, which is insufficient. Besides, the main subject of the work doesn't fit very well with the subject of the journal (biosensors), so a great amount of supplementary work is necessary to be done in order to make the manuscript suitable for publication.

Based on these consideration, this manuscript cannot be accepted in this form for publication, and another journal from MDPI maybe is more suitable.

Reviewer 3 Report

In this manuscript, the authors described an extended Smoluchowski rate equation coupled with a fragmentation scheme to investigate the growth of size-selected silver NCs generated via a modulated pulsed power magnetron sputtering. In addition, the combined experimental and simulation results suggest that as peak power Pp increases, the formation of lager neutral NCs and the loss of total current intensity of anion and cation are mainly due to the recombination of cation and anion species during condensation. I think the manuscript is well organized as well as presents sufficient data to support the authors’ hypothesis. Therefore, I recommend the acceptance of this manuscript after accommodating the following minor comments.

Comment 1. In Figure 2, the authors need to add a detailed explanation of the data in the caption even though it is described in the main text. For example, red dots and black line means experimental data and fitting curve, respectively.

Comment 2. Although it is too obvious, it is recommended to add EDS data to verify that the manufactured particles are composed of silver.

Reviewer 4 Report

The author shows an very interesting way to generate silver nanoparticles which have potential interests in sensing systems using LSPR and SERS mechanism. Theoritical and experimental work is well explained and the proposed method can be useful for reasearches as it improves and explains a concrete method for silver nanoparticles generation. Higher details (or maybe) a concrete apllication in biosensing could be included in the paper to be more in line with the special issue, but in my opinion the technique proposed as well as results deserve to be publish.

Round 2

Reviewer 2 Report

I am pleased to see that all issues were addressed by the author, so now the manuscript can be accepted for publication.